

# Generative invertible quantum neural networks

### Armand Rousselot[1*] and Michael Spannowsky[2]

**1** Interdisciplinary Center for Scientific Computing, Universität Heidelberg, Germany
**2** Institute for Particle Physics Phenomenology, Department of Physics,
Durham University, DH1 3LE, United Kingdom

* armand.rousselot@iwr.uni-heidelberg.de

## Abstract

**Invertible Neural Networks (INN) have become established tools for the simulation and generation of highly complex data. We propose a quantum-gate algorithm for a Quantum Invertible Neural Network (QINN) and apply it to the LHC data of jet-associated production of a Z-boson that decays into leptons, a standard candle process for particle collider precision measurements. We compare the QINN's performance for different loss functions and training scenarios. For this task, we find that a hybrid QINN matches the performance of a significantly larger purely classical INN in learning and generating complex data.**

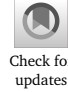

# 1 Introduction

Generative modelling has been a field of particular interest in machine learning research, being vastly improved by successful model architectures, including Variational Autoencoders (VAE), Generative Adversarial Networks (GAN) and Invertible Neural Networks (INN) [1–3]. Among other applications, their use in event generation has been extensively investigated [4–6]. Their advantages over the Markov chain Monte Carlo (MCMC) techniques [7–11], which had so far established themselves as the leading LHC simulation and interpretation methods go beyond an increase of inference speed. Furthermore, generative models can be trained end-to-end, allowing for a much more comprehensive range of applications such as unfolding [12–14], anomaly detection [15–19] and many more [20].

However, the large parameter space of these Neural Networks (NN), which allows them to model complex interactions, also leads to a massive demand for computing resources. The size of popular NN architectures has long reached the boundary of computational feasibility. Quantum Machine Learning (QML) introduces the power of quantum computing to the existing foundation of machine learning to establish and then exploit the quantum advantage for a performance increase exclusive to quantum algorithms. While gate-based quantum computing differs significantly from classical computing, many equivalents to aforementioned classical generative networks have already been constructed, including Quantum Autoencoders [21] and Quantum GANs [22–27]. The notable exception is INNs [28,29], which have not yet been transferred to the realm of QML. Such networks would be a desirable addition to the array of Quantum Neural Networks (QNN). While tractability of the Jacobian determinant in classical INNs enables them to perform density estimation, which intrinsically prevents mode collapse, the full Jacobian matrix can usually not be computed efficiently [30]. A fully tractable Jacobian in INNs, available for QNNs, would allow efficient learning of the principal data manifolds [31–34], opening up opportunities for interpretable representation learning and new insights into underlying processes.

Coupling-based INN architectures have empirically shown to be more resilient to the vanishing-gradient problem [28], which lets them directly benefit from deep architectures with many parameters. However, many of the INN applications listed so far already require considerable resources for training. Current research suggests that quantum models could circumvent this need for an immense parameter space. They outclass regular NNs in terms of expressivity, being able to represent the same transformations with substantially fewer parameters [35–39]. This theoretical groundwork is bolstered by several instances of specifically constructed QML circuits presenting significantly more efficient solutions than classically possible to specially designed problems [40–43]. QNNs have already been successfully applied to relatively limited high-energy physics problems [21,25,44–46,46–51], along non-QML approaches [52–56]. However, to our knowledge, there has not yet been an attempt to construct an invertible QNN that can be used as a density estimator through its invertibility for generative tasks.

With this work, we aim to fill the remaining gap of a quantum equivalent to classical INNs, developing a Quantum Invertible Neural Network (QINN). We show how each step in the QNN pipeline can be designed to be invertible and showcase the ability of a simulated network to estimate the density of distributions. As a proof-of-principle, we apply our model to complex simulated LHC data for one of the most important and most studied high-energy physics processes,

$$pp \rightarrow Zj \rightarrow \ell^+ \ell^- j \,,$$

and show its capability to reconstruct the invariant mass $M_Z$. While currently available noisy intermediate-scale quantum computers (NISQ) cannot support models required for even basic generative tasks, the concept of inverting QNNs still shows promise in light of the aforemen-

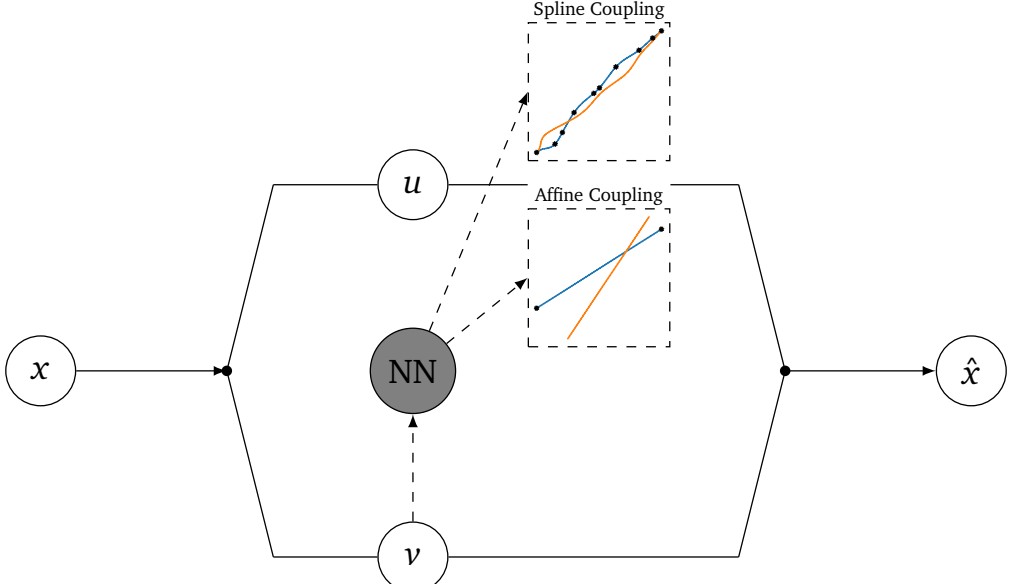

Figure 1: Layout of a coupling block. The input is split into two halves along the feature dimension. One half is used as neural network input to predict parameters for either an affine or a spline transformation of the other. In the end, both halves are fused again. The blue lines illustrate example coupling transformations of both block types, based on the network output (black knots). The orange line shows the inverse transformation.

tioned theoretical and simulation-based results, documenting their increased expressivity. As we will confirm in our experiments, QINNs have the potential to express the same set of transformations as classical INNs with much fewer parameters.

This study is structured as follows: In Sec. 2 we provide a short review of classical Invertible Neural Networks. Then, Sec. 3 is dedicated to our proposal for a Quantum Invertible Neural Network based on quantum-gate algorithms and outlining its technical challenges. Next, we apply the QINN to simulated LHC data in Sec. 4. We offer a brief summary and conclusions

## 2 Classical invertible neural networks

To illustrate the advantages of invertible models and to benchmark the performance of the QINN, we first present the architecture of a classical INN. As we will see in the following section, we can train any model to perform density estimation as long as it fulfils the Jacobian determinant's requirements of invertibility and availability. To meet these requirements, the INNs are constructed using coupling blocks, see Fig. 1. Each coupling block splits the data vector in two halves $x = [u, v]^T$ along the feature dimension, transforming one half at a time. The parameters for this transformation are predicted by a small model, e.g. a neural network, based on the other half of the data. The transformation used as a benchmark in this work are cubic spline coupling blocks [57]. However, we will introduce affine coupling blocks first as a simpler example.

The affine coupling block performs an element-wise linear transformation on one half $u$, with parameters $s(v), t(v)$ predicted by a neural network from the other half $v$

$$\begin{bmatrix} \hat{u} \\ \hat{v} \end{bmatrix} = \begin{bmatrix} u \odot e^{s(v)} + t(v) \\ v \end{bmatrix}. \tag{1}$$

The inverse is now trivially given by

$$\begin{bmatrix} u \\ v \end{bmatrix} = \begin{bmatrix} (\hat{u} - t(\hat{v})) \odot e^{-s(\hat{v})} \\ \hat{v} \end{bmatrix}. \tag{2}$$

In each layer the Jacobian is an upper triangular matrix, as the transformation of $u_i$ only depends on $v$ and $u_i$

$$J = \begin{pmatrix} \text{diag } e^{s(v)} & \text{finite} \\ 0 & \mathbb{I} \end{pmatrix}. \tag{3}$$

Therefore the Jacobian determinant can be calculated and using $\det J(f_1 \circ f_2) = \det J(f_1) \det J(f_2)$, the Jacobian determinant for the entire network can be assembled from all blocks individually. In this configuration of the coupling block, the lower half $v$ stays unchanged. After each block, a unitary soft-permutation matrix is applied to the data vector to mitigate this fact. Since this matrix is invertible and the Jacobian determinant is 1, this does not influence the network prerequisites.

The spline coupling block is functionally similar, except that the transformation performed in each block is a cubic spline of $u$, predicted from $v$. Given several buckets, $\xi - 1$ (hyperparameter), the coupling block subnetwork predicts $2\xi + 2$ parameters for each $u_i$. These parameters serve as anchor points for a cubic spline on $u_i$, meaning $x, y$ coordinates for $\xi$ bucket boundaries (ensuring that $x_i < x_{i+1}, y_i < y_{i+1}$) and two values to define the slope of the interpolating spline in the two outer anchor points. Since each bucket is simply a monotonous cubic function of $u$, the whole transformation is invertible, and the same argument for the triangularity of the Jacobian from the affine coupling blocks still holds. Spline couplings trade increased computation time for higher expressivity per parameter, which is why we choose them for the comparison to QINNs in Sec. 4. We implement these networks in PYTORCH using the FREIA module [58], where finer implementation details can be found.

## 2.1 Density estimation using invertible models

We aim to train the model to estimate the density of some given training data with distribution $p(x)$. This is achieved by learning a bijective mapping $f$, defined by network parameters $\theta$, from some desired distribution $p(x)$, $x \in R^n$ to a normal distribution

$$f(x \sim p(x)|\theta) \sim \mathcal{N}^n_{0,1}(z). \tag{4}$$

With an invertible network function $f$, new data can straightforwardly be generated from the desired distribution by sampling $x \sim f^{-1}(z \sim \mathcal{N}^n_{0,1}(z)|\theta) =: p(x|\theta)$. To construct a loss function, we can use the change of variables formula

$$p(x|\theta) = \mathcal{N}^n_{0,1}(f(x|\theta)) \left| \det\left(\frac{\partial f}{\partial x}\right) \right|. \tag{5}$$

The training objective is to find the network parameters $\theta$ which maximise the probability of observing the training data from our model $f$

$$\max_{\theta} p(\theta|x) \propto p(x|\theta)p(\theta). \tag{6}$$

We can transform this expression to obtain a loss function, by minimizing the negative log likelihood, and substitute Equation 5 for $p(x|\theta)$. Finally we assume a gaussian prior on $p(\theta) = \exp(\tau\theta^2)$[1] and write $J = \det\left(\frac{\partial f}{\partial x}\right)$ to get the loss function

$$\mathcal{L} = \mathbb{E}\left[ \frac{||f(x|\theta)||_2^2}{2} - \log|J| \right] + \tau||\theta||_2^2. \tag{7}$$

---

[1]In our experiments we found it sufficient to set $\tau = 0$ for quantum parameters, which indicates a uniform weight prior on the qubit rotation angles.

Therefore all we need to train a model to perform density estimation is invertibility and the ability to calculate its Jacobian determinant w.r.t. the input $x$.

## 2.2 Maximum mean discrepancy

We can improve targeted observables by using a so-called Maximum Mean Discrepancy (MMD) loss [59]. MMD estimates the distance of two given distributions in some Reproducing Kernel Hilbert Space. In our case we want to minimize the distance $d(p(\phi(x)), p(\phi(f^{-1}(z|\theta))))$ given some features of our choice $\phi$. MMD approximates this distance given samples from the two distributions $X \sim p(x)$ and $Y \sim p(f^{-1}(z|\theta))$ and a function $\phi$

$$
\begin{aligned}
\mathscr{L}_{MMD}^2 &= \left\| \frac{1}{|X|} \sum_{x \in X} \phi(x) - \frac{1}{|Y|} \sum_{y \in Y} \phi(y) \right\|_2^2 \\
&= \frac{1}{|X|^2} \sum_{x \in X} \sum_{x' \in X} \phi(x)^T \phi(x') + \frac{1}{|Y|^2} \sum_{y \in Y} \sum_{y' \in Y} \phi(y)^T \phi(y') - 2 \frac{1}{|X||Y|} \sum_{x \in X} \sum_{y \in Y} \phi(x)^T \phi(y).
\end{aligned}
\tag{8}
$$

Since all appearances of $\phi$ involve the inner product $\phi^T(\cdot)\phi(\cdot)$ we can use the kernel trick to substitute them with a matching kernel $k$ that calculates the induced vector product $<\cdot, \cdot>_\phi$

$$
\begin{aligned}
\mathscr{L}_{MMD}^2 &= <X, X>_\phi + <Y, Y>_\phi - 2 <X, Y>_\phi \\
&= \overline{k(X,X)} + \overline{k(Y,Y)} - 2\overline{k(X,Y)}.
\end{aligned}
\tag{9}
$$

The kernel should be selected according to the distributions that are being compared. Since a Gaussian-like distribution is generally a sufficient approximation in most cases, we use the gaussian kernel function

$$
k_{\text{gauss}}(x, y) = \exp\left( -\frac{||x - y||_2^2}{2\sigma^2} \right).
\tag{10}
$$

In our experiments we will also apply a MMD to the invariant mass of a $Z$-Boson $M_Z$. In this case, since it has a Breit-Wigner distribution we use the corresponding kernel

$$
k_{\text{Breit-Wigner}}(x, y) = \frac{\sigma^2}{\sigma^2 + ||x - y||_2^2}.
\tag{11}
$$

The parameter $\sigma$ determines the width of the kernel. Choosing this width correctly is often more important than the correct kernel shape. If $\sigma \ll ||x - y||_2$ then $k(x, y) \simeq 0$ and if $\sigma \gg ||x - y||_2$ then $k(x, y) \simeq 1$. In both cases, $\mathscr{L}_{MMD}$ will be very close to zero, and the gradient will vanish. While methods exist that adaptively change $\sigma$ over the training to fit the model distribution, another easy and effective way is to use multiple kernels of different widths.

Classical INNs generally do not benefit greatly from an additional MMD loss for the overall training apart from improving specifically targeted observables. However, we found that applying MMD on both the latent and input sides of the QINN to all training observables, even simultaneously to train a gaussian latent and the target input distribution, significantly improves performance.

## 3 Invertible quantum neural networks

As we will illustrate in this section, QNNs lend themselves well to being inverted, requiring very few modifications from their most basic form. For example, the underlying architecture of

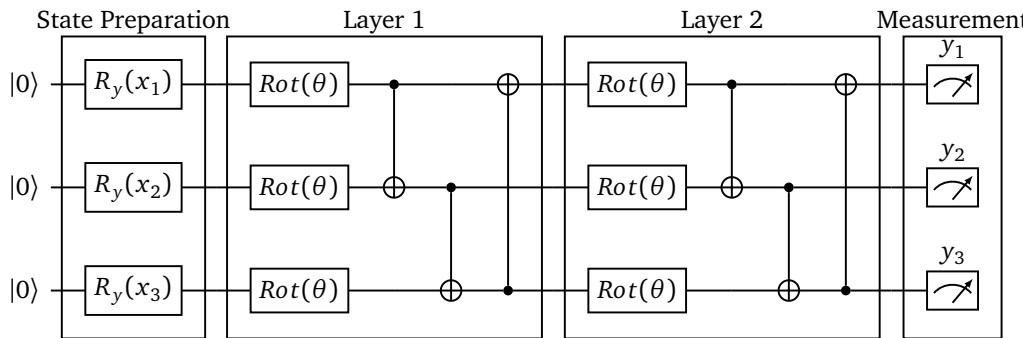

Figure 2: An overview of the model circuit. We show a three-qubit, two-layer model following the hardware efficient ansatz from [61]. The state preparation uses angle encoding, where each feature is encoded on its qubit. The learnable parameters of the model circuit are the rotation angles $\theta$ in each layer.

a circuit-centric QNN [60] can be split into three distinct parts, *state preparation, model circuit* and *measurement* as seen in Fig. 2.

State preparation transforms a given data point $x = (x_1, \ldots, x_n)^T$ from the classical domain into a quantum state $|x\rangle$. One of the simplest methods to achieve this is angle encoding, in which each input dimension is interpreted as an angle for a qubit rotation. Therefore, the number of qubits is fixed as the number of feature dimensions $n$. The entire dataset is first scaled to $\mathbb{R}^n \rightarrow [0, \pi]^n$. Next, we apply a global linear transformation to the input with trainable parameters $a \in [0, 1]^n$; $b \in [0, 1-a]^n$, clipped to prevent $x \notin [0, \pi]$. We obtain a quantum state by defining a state preparation operator $S_x = R_y(x)$, which acts on the initial state

$$|x\rangle = S_x |0\rangle^{\otimes n} = \bigotimes_{i=1}^{n} \cos(x_i) |0\rangle + \sin(x_i) |1\rangle \,. \tag{12}$$

When performed this way, inverting the state preparation is straightforward since simply measuring $P(\text{qubit } i = 1)$ gives $\langle x|\sigma_{z,i}|x\rangle = \sin^2(x_i) \implies \arcsin\left(\sqrt{\langle x|\sigma_{z,i}|x\rangle}\right) = x_i$.

The model circuit is the quantum analogue of a classical neural network, mapping the prepared state to the output state $|x\rangle \mapsto U|x\rangle =: |y\rangle$. The circuit comprises modular layers $U = U_l \ldots U_1$. Each layer starts with a rotation gate for each qubit $Rot(\phi, \theta, \eta)$ parameterized by trainable weights [61]. Afterwards, the qubits are entangled by applying CNOT gates between each adjacent pair of qubits and from the last to the first qubit. The entire model circuit can be straightforwardly inverted by applying the adjoint to each layer in the reverse order

$$U^\dagger = (U_l \ldots U_1)^\dagger = U_1^\dagger \ldots U_l^\dagger \,. \tag{13}$$

The adjoint operation to each of the gates is simple to obtain

$$Rot^\dagger(\phi, \theta, \eta) = Rot(-\eta, -\theta, -\phi)\,, \tag{14}$$

$$CNOT^\dagger(i, j) = CNOT(i, j)\,. \tag{15}$$

Finally we measure $P(\text{qubit } i = 1)$ for all $n$ qubits and apply another trainable global linear transformation with parameters $c, d \in \mathbb{R}^n$

$$|y\rangle \mapsto c \langle y|\sigma_z|y\rangle + d\,. \tag{16}$$

Inverting the final measurement is not directly possible, as many different states can lead to the same expectation value $\mathbb{E}[y]$. Note that the model function can still be bijective if no two created states yield the same $\mathbb{E}[y]$. Since the network input is $x \in R^n$ the set of created final

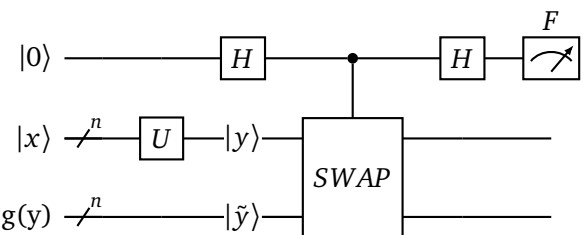

Figure 3: The SWAP-test, comparing a state $|y\rangle = U|x\rangle$ created in the forward direction to $|\tilde{y}\rangle = g(y)$ created by the ISP. The CSWAP gate acts pairwise on the wires, i.e. $y_1$ is swapped with $\tilde{y}_1$, etc.

states only exists in a $S \subseteq \mathbb{C}^{2^n}$, $s.t. \dim S = n$ subspace of the state space. However, we need to ensure that $S$ does not share any $\mathbb{E}[y]$, as well as find the proper method to perform the inverse state preparation (ISP) for each data point.

## 3.1 Inverse state preparation

Given a model circuit $U$ and a data point $y$ that arose from measuring $|y\rangle = U|x\rangle$, it is infeasible to search for an ISP method that creates $|y\rangle$ from $y$. Therefore we instead aim to train the model circuit $U$ in a way such that for a given fixed ISP $g$, the state $|y\rangle$ before the measurement and $|\tilde{y}\rangle := g(\langle y|\sigma_z|y\rangle)$ are as close as possible. We evaluate the fidelity, measuring the "closeness" of two quantum states [62],

$$F = \langle \tilde{y}|y\rangle , \tag{17}$$

using the SWAP-Test shown in Fig. 3. Which side of the model we perform the SWAP-Test on does not matter, as the operator $U$ is unitary. We train the entire model, the model circuit and all ISP parameters, to adhere to $F \simeq 1$ for the loss function

$$\mathscr{L}_F = \lambda_F(\log(F)). \tag{18}$$

While the model is invertible if the fidelity $F \simeq 1$, the opposite is not necessarily true. In fact for a given circuit $U$ we can find exponentially many different states $|\tilde{y}\rangle$ such that

$$\tilde{x} := \langle \tilde{y}|U\sigma_z^{\otimes n}U^{\dagger}|\tilde{y}\rangle = x. \tag{19}$$

Thus, it seems more natural to define the invertibility loss directly on $\tilde{x}$. We construct an alternative loss function which only trains the model to adhere to $\tilde{x} \simeq x$

$$\mathscr{L}_{MSE} = \lambda_{MSE}(x - \tilde{x})^2. \tag{20}$$

We compare both loss functions quantitatively in Sec. 4.

While one can select any ISP method of one's choosing, a fixed ISP will often be too restrictive in practice. We, therefore, allow the model to learn its ISP by creating a separate module which is only called in the inverse direction, mapping a measurement $y$ to the quantum state $|y\rangle = g(y)$, see Fig. 4. This module combines a small classical neural network $g_C$ with a quantum neural network $g_Q$. First, the neural network predicts $3n$ angles $\psi$ from $y$

$$y \xmapsto{g_C} \psi \in \mathbb{R}^{3n} , \tag{21}$$

which serve as inputs for the $Rot$ gates. The quantum state prepared in this way is then further transformed by the quantum neural network to create the input for the (inverse) model circuit

$$\psi \xmapsto{Rot} |\psi\rangle \xmapsto{g_Q} |\tilde{y}\rangle . \tag{22}$$

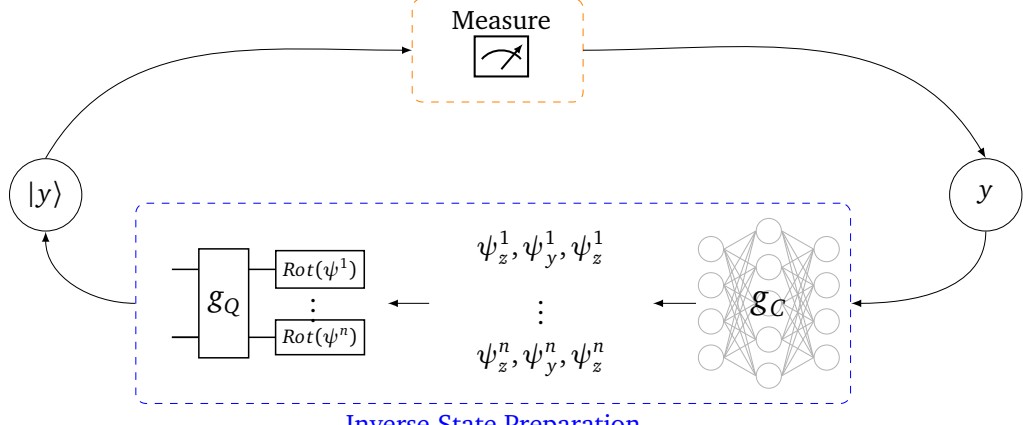

Figure 4: A diagram of the learnable ISP method. To map a measurement back to a quantum state, first a NN predicts $3n$ angles, which serve as state preparation for $|\psi\rangle$, which is then transformed further by a separate QNN that recreates $|y\rangle$.

With these additional steps, we can ensure that the model is trained to be invertible. Furthermore, as explained in Sec. 2.1, the model's Jacobian needs to be tractable for density estimation. For the QINN, it can be obtained in a similar way that the gradients are calculated by using parameter shift rules [63, 64].

The potential of a QINN is twofold. Firstly, a QINN provides the ability to compute the full Jacobian matrix. Unlike classical INNs, which only allow for efficient computation of the Jacobian determinant [30], a full Jacobian would open up opportunities for a new array of applications. There has been extensive research into efficient learning of the principal data manifolds in distributions [31–34], yet it remains a challenging and computationally expensive task in higher dimensions. Exploiting the full Jacobian of a QINN to encode the principal manifolds in selected latent variables would come at very little additional cost. This advantage of QINNs could pave the way towards learning interpretable representations and new insights into underlying processes.

The second advantage of a QINN lies in the increased expressive power provided by quantum models, which extensive theoretical work has documented [35–38]. There has been considerable effort to define quantum equivalents of multiple generative models in the current machine learning landscape [65]. Sometimes, simulation of distributions created by quantum circuits is not efficiently possible with classical methods [66]. For example, the authors of [67] show an advantage in learning density estimation via samples with Quantum Circuit Born Machines [68] using Clifford group gates [69]. Even though QINNs operate fundamentally differently, since we marginalize over the measured state distribution, there remains reason to assume increased expressivity of a QINN over the classical counterpart, which we aim to further establish by the experiments shown in Sec. 4.

# 4 Application to high energy physics

We evaluate the performance of the QINN by learning to generate events of the LHC process

$$pp \rightarrow Zj \rightarrow \ell^+\ell^- j\,. \tag{23}$$

We simulate 100k events using MADGRAPH5 [70] with generator-level cuts for the transverse momentum and the rapidity of the leptons of 15 GeV $< p_{T,\ell^\pm} <$ 150 GeV and $\eta_{\ell^\pm} <$ 4.5 as

Table 1: Hyperparameters for the QINN and INN used for training and setup. The hyperparameters for the INN were chosen such that the performance of the QINN and the INN were comparable while keeping the number of trainable parameters as low as possible.

| Hyperparameter | QINN | INN |
|---|---|---|
| LR scheduling | 1cycle [73] | same |
| Maximum LR | $10^{-3}$ | same |
| Start/Final LR factors | $4 \cdot 10^{-2}/10^{-4}$ | same |
| Epochs | 200 | same |
| Batch size | 128 | same |
| ADAM $\beta_1$, $\beta_2$ | 0.9, 0.9 | 0.5, 0.9 |
| # Spline bins | - | 5/8/8 |
| # Coupling blocks | - | 6/9/10 |
| Layers per block | - | 3/3/3 |
| Hidden dimension | - | 8/12/24 |
| # Forward quantum layers | 12 | - |
| # ISP quantum layers | 8 | - |
| # ISP NN layers | 3 | - |
| # ISP hidden dimension | 32 | - |
| $\lambda_{MMD}$ (input/latent) | 1.0/0.5 | - |
| $\lambda_{M_Z}$ | 0.375 | 0.5 |
| $\lambda_{F/MSE}$ | 10.0 | - |
| MMD kernel widths | [0.02, 0.1, 0.4, 1, 5] | same |
| # Trainable parameters | 2k (QNN $\sim$ 300, NN $\sim$ 1.7k) | 2k/6k/16k |

well as the energy of the $Z$-Boson $E_Z < 1900$ GeV. The data sample is split into a training, a validation and a test sample following the split of 37.5%/12.5%/50%.

We compare the two methods of training invertibility described in Sec. 3.1, the fidelity of the quantum states and a mean squared error. Finally, we train a classical INN with the spline coupling block architecture presented in Sec. 2 and compare the performance based on the number of model parameters required. The setup of all models and hyperparameters can be found in Table 1. We implement the training pipeline with PYTORCH [71], where we use PENNYLANE [72] to implement the QINN and FREIA [58] for the classical INN. We train all networks on the observables $p_{T,\ell^\pm}, \Delta\phi_{\ell\ell}, \eta_{\ell^\pm}$, which we can use to reconstruct the $Z$-Boson. The models are trained for 200 epochs with an MMD loss as described in Sec. 2.2 on the $M_Z$ distribution, which significantly improves the results in this observable for all models. The MMD loss on input and latent observables for the QINN are used throughout the entire training process.

## 4.1 Comparing fidelity and mean squared error for the loss function

To decide which of the loss functions for invertibility is more advantageous for the QINN, we perform experiments with the same hyperparameters, only changing the loss. A comparison of the results is shown in Fig. 5. While both results are similar in performance of $M_Z$, the fidelity loss creates significant artefacts in the $\Delta R_{\ell^+,\ell^-}$ distribution whenever we use the MMD loss necessary to improve $M_Z$. This equations with the intuition of the MSE loss allowing for a more flexible choice of ISP. We, therefore, proceed with the MSE loss for invertibility throughout the rest of this work.

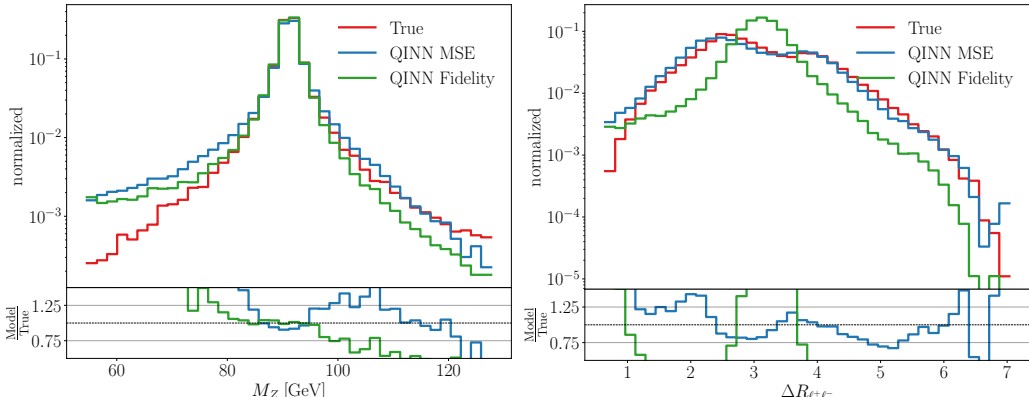

Figure 5: $M_Z$ and $\Delta R_{\ell^+,\ell^-}$ of the $Z$ reconstructed from the leptons as generated by the QINN with Fidelity and MSE loss for invertibility. True shows the distribution of the entire test set.

## 4.2 Classical INN versus quantum INN

We choose three INN sizes: 2k parameters to match the QINN in parameter number, 6k and 16k parameters to approximately lower- and upper-bound the QINN performance. In Fig. 6, we first compare correlations of low-level observables between the QINN and the 16k INN. While both networks cannot learn the $p_x$ double peak structure due to their limited size, they both show no problems learning the hard $p_T$ cuts. Furthermore, the QINN shows no additional signs of deterioration or artefacts in the low-level observables that may have arisen from underparameterization apart from the ones also present in the INN.

The networks' ability to capture high-dimensional correlations can be tested by reconstructing the $Z$-Boson observables, specifically the invariant mass $M_Z$ from the generated leptons. We show these reconstructed results in Fig. 7. It is immediately apparent that the 2k parameter INN is nowhere as expressive as the QINN. In fact, the QINN even outperforms the 16k parameter INN at reconstructing the sharp peak of the $M_Z$ distribution, though it does not match the tails of the shown distributions as well as the 16k INN. Comparing the QINN to the 6k INN, it arguably even outperforms a classical network three times its size. With an average deviation of the reconstructed observables of $||\frac{\tilde{x}-x}{x}|| < 2.1\%$, we can also determine that the MMD loss does not dominate the optimization process and the QINN does learn to perform an invertible transformation.

In conclusion, we find the performance of the QINN to be equivalent to that of a classical INN with around $3-8$ times the number of parameters on this 5 dim task, with most of the QINN parameter count still being attributed to the classical NN.

## 5 Conclusion

Generative modelling has become increasingly important for simulating complex data in various science areas. In recent years, neural network techniques, such as Variational Autoencoders, Generative Adversarial Networks and Invertible Neural Networks, have received attention, showing promising outcomes. At the same time, algorithms designed for quantum computers have opened a new avenue to expand on existing classical neural network structures. For example, quantum-gate algorithm-based Quantum Variational Autoencoders and Quantum Generative Adversarial Networks have been studied thoroughly. They have been shown empirically to match or even outperform their classical counterparts on specific tasks

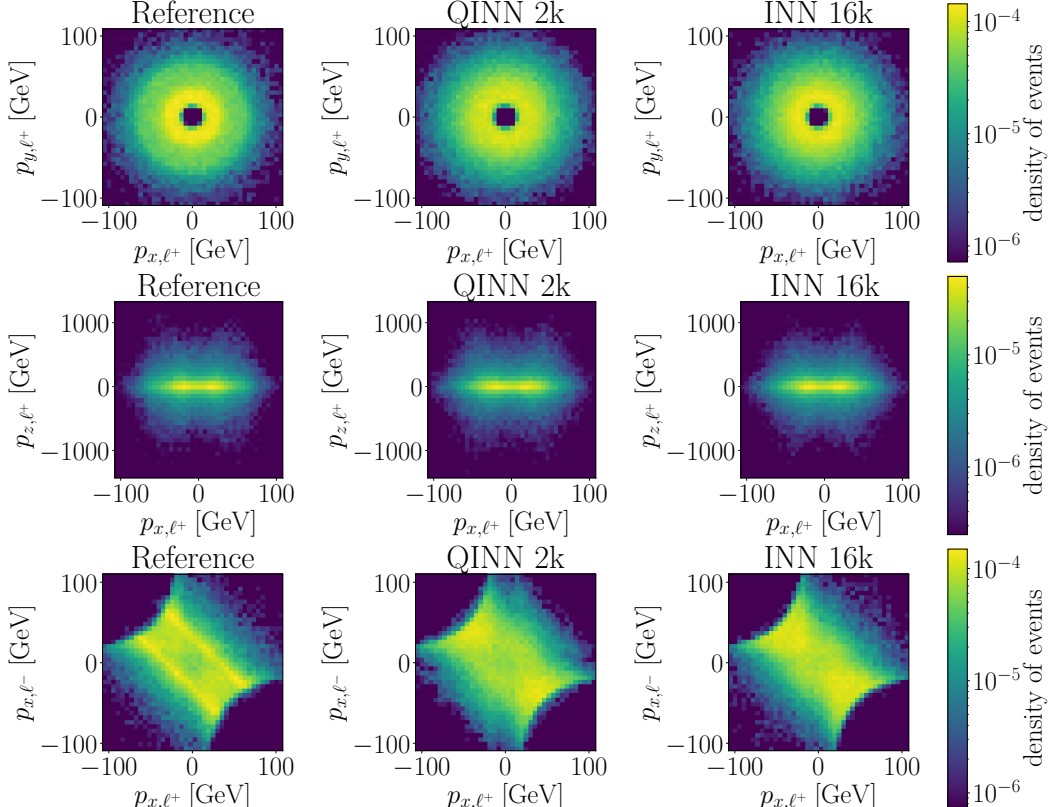

Figure 6: 2d correlations of selected $\ell^{\pm}$ observables. We show the distribution in the dataset (left) the one generated by the QINN (center) and the one generated by the 16k parameter INN (right).

or when limiting the size of the classical network, thereby indicating that QNNs can offer a larger expressivity or faster and more robust network training.

In this work, we proposed a novel approach for Quantum Invertible Neural Networks and highlighted their use as density estimators in generative tasks. By applying the QINN to the simulation of final states of the LHC process $pp \rightarrow Zj \rightarrow \ell^+\ell^-j$, we showed its ability to reconstruct the $Z$-Boson with significantly fewer parameters than classical INNs. Our model combines the conventional QNN architecture, consisting of the trinity of state preparation variational quantum circuit and measurement, with a classical-quantum hybrid network for learning an Inverse State Preparation. Furthermore, we demonstrate how the combined model can be trained to invert the quantum measurement of the QINN, allowing for a reversible transformation. Through the property of having the entire network Jacobian at one's disposal, performing density estimation with QNNs could lead to new insights and better understanding of the modelled generative processes.

The hybrid QINN with 2k trainable parameters, most of which originate in the classical network part, showed to be more expressive than its entirely classical counterpart, thereby evidencing a gain in expressivity due to the inclusion of the quantum circuit. This encouraging result motivates the detailed future study and employment of QINNs in complex generative tasks.

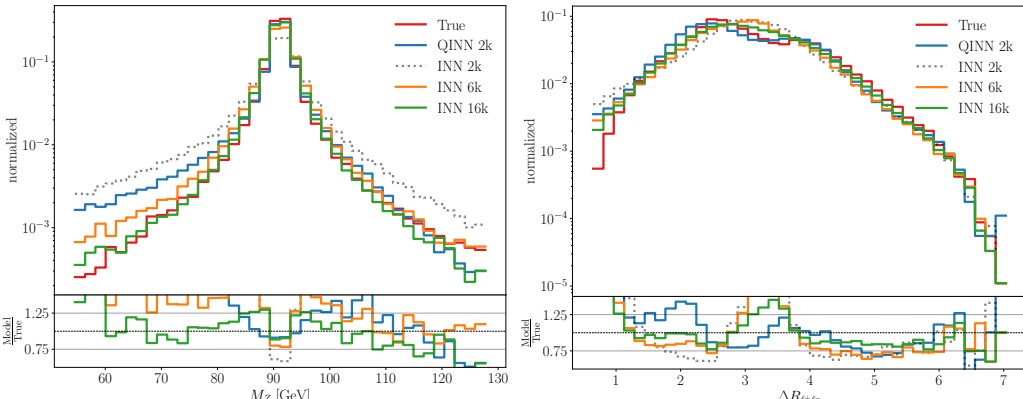

Figure 7: $M_Z$ and $\Delta R_{\ell^+,\ell^-}$ of the $Z$ reconstructed from the leptons as generated by the QINN and the reference INNs. True shows the distribution of the entire test set.

## Acknowledgements

We thank Tilman Plehn for valuable discussions and encouragement during this project.

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
