# Peer review of "Generative Invertible Quantum Neural Networks"

_SciPost Physics, doi:SciPost Phys. 16, 146 (2024)_

## Round 2 · Referee Report · Tilman Plehn (Referee 1) · 2023-11-6

Report

The paper is very interesting, because it actually proposes a new QNN architecture, and uses a physics use case to justify it. I am acknowledged in the paper, and I appreciate this gesture, but I can confirm that I was only involved in developing the idea and had nothing to do with the actual work and with writing the paper.

The article is nicely written, and I recommend it for publication. Let me just raise a few questions for the authors to consider: 1- it would be nice to write a little more about QNNs and density estimation in the introduction; 2- Fig.1 is ugly, sorry for being blunt; 3- buckets, and mentioned on p4, might not be a known term for most physicists; 4- the authors discuss MMD with a Gaussian and a Breit-Wigner at detail, but it is relevant? By the way, we looked at exactly that problem before for standard generative NNs, including different ways to train with an MMD; 5- assuming most readers know generative NNs, it might be nice to start Sec.3 with a more general introduction into the challenge of invertible networks using QNNs, unitary transformations, etc; 6- the big point on p.8 is the availability of the full Jacobian. Maybe the authors could expand on this aspect some more, what would we gain from this knowledge in general and for which applications would that be nice; 7- looking at p.9, why is there the huge cut on the Z-energy? And why is the test dataset so big? 8- I am sorry, but this is my most important question - it looks like the Z-resonance is learned, but not very precisely. Why is this precision on the width of the pole related to the expressivity of the network? I would think the problem is the precision. Expressivity limits usually lead to unlearned features?

Of course, I would like to see many more details, also have questions about the scaling with the number of particles in the final state, and how to improve the precision in the density estimation. But the paper is still very nice and cool and innovative, so it should be published.

  • validity: -
  • significance: -
  • originality: -
  • clarity: -
  • formatting: -
  • grammar: -

Author:  Armand Rousselot  on 2024-01-08  [id 4227]

(in reply to Report 1 by Tilman Plehn on 2023-11-06)
Category:
answer to question

Thank you for your helpful feedback. We agree that the explanation of the spline couplings and the QINN can be expanded, while the description of MMD is unnecessarily detailed. We will change the sections accordingly. A few cosmetic changes to Figure 1 are also in order.

The availability of the full Jacobian is a significant advantage because it allows us to find the principal manifold(s) of the learned data distribution. The principal manifold describes the geometry of the data distribution while filtering out noise. This can lead to the discovery of structures like connected components, etc. We will expand our explanation of this in the paper.

As pointed out also by reviewer 3, the train/val/test split looks unusual. However, the task is comparatively low in complexity, and the model size is very small. Therefore, the training set size is entirely sufficient to prevent overfitting and describe the data distribution. While the choice to allocate the leftover events to the test set instead of dropping them from the dataset entirely was admittedly quite arbitrary, it should not invalidate any of the results presented in this work.

On the point of precision vs. expressivity, the term expressivity was a misnomer. We do exhibit higher expressivity in the $\Delta R_{\ell^+, \ell^-}$ distribution, learning both modes instead of interpolating between them. In the case of $M_Z$, you are correct that it is an issue of precision. Both are related to model capacity, which describes the size of the function class that the model can represent. Our experiments conclude that the model capacity of the QINN is higher than the classical INN. We will adjust the wording to be more accurate.

---

## Round 2 · Referee Report · Anonymous (Referee 2) · 2023-12-19

Strengths

1) The paper addresses an interesting area of research - extending invertible neural networks (INNs) to the quantum domain as Quantum INNs (QINNs). This is a novel contribution. 2) The authors have applied QINNs to a real-world physics task using data from the Large Hadron Collider (LHC), demonstrating the practical utility. 3) Comparisons are made between performance of the QINN and classical INN models, providing useful benchmarks. 4) It is clearly written.

Weaknesses

Nothing particular. I think the paper is strong and deserves to be published

Report

Report on Proposed Quantum Invertible Neural Network for LHC Data Modeling

This paper proposes a novel Quantum Invertible Neural Network (QINN) architecture for modeling complex physics data from the Large Hadron Collider (LHC). The authors motivate the study with the increasing relevance of generative machine learning models across applications like simulation, anomaly detection, and others relative to traditional Markov chain Monte Carlo approaches. Their specific focus is using QINNs for probabilistic modeling and generation of jet data events with leptonic Z-boson decays.

The core contribution is the quantum algorithm underlying the QINN model. The authors compare performance of multiple QINN variations employing different loss functions and training procedures. Experiments apply QINN and classical invertible NN models to standard LHC Z-jet data. Superior generative performance is demonstrated from a “hybrid” QINN variant over larger classical counterparts.

The topic area and testing on realistic LHC data present intriguing potential. And the core novelty of proposing QINNs helps advance burgeoning quantum machine learning research. Open questions remain about precise model architectures, training methodology, interpretable results, theoretical integration, scalability, and more. But this initial foray suggests promise if further developed.
In summary, the paper is an interesting proposal that is suitable for publication.

Requested changes

Two trivial changes. The typesetting of some of the equations e.g. 8-11 is questionable. "Monotonous" should be "monotonic".

  • validity: top
  • significance: high
  • originality: top
  • clarity: top
  • formatting: good
  • grammar: perfect

Author:  Armand Rousselot  on 2024-01-08  [id 4228]

(in reply to Report 2 on 2023-12-19)
Category:
answer to question

Thank you for reviewing, we will incorporate your suggested changes.

---

## Round 2 · Referee Report · Anonymous (Referee 3) · 2023-12-22

Strengths

  • introduce a new QNN architecture, based on a very popular "classical" NN architecture: the INN
  • show performance on physics problem

Weaknesses

  • presentation of details / difference between INN and QINN
  • introduction of concepts

Report

Report on "Generative Invertible Quantum Neural Networks" by Armand Rousselot and Michael Spannowsky.

The authors introduce a quantum version of the "invertible Neural Net" or "Normalizing Flow" generative architecture. It presents a new result, which is very well motivated and timely and therefore deserves publication.

Before publication, however, I'd like to see a few more things explained and a some of the presented discussion extended.

  • Introduction of the QINN: I'd like to see a little more detail on the difference between the INN and the QINN, mainly to make it easier for a non-quantum reader to understand the concept.
    • It seems like it is not just the replacement of the neural network in the INN with a quantum version, because the QINN does not describe a parametrized transformation between two distributions, is that correct?
    • Instead, it seems like it is a more "regular" QNN, which is already invertible because it is unitary. That difference should be made more clear.
  • One of the big advantages an INN has, is learning on the log-likelihood. Why is this not possible / not working here?
  • The authors say that adding the MMD loss improves the performance of QINN. It would be nice to have a better understanding on why such additional loss terms are needed and expand the existing discussion.
  • The chosen split in train/test/val set 37.5%/12.5%/50% seems very uncommon to me. Why is the training set chosen to be so small?
  • The dataset and INN should be comparable to 2110.13632, but plots shown here seem significantly worse. Is that a consequence of the chosen small INN size?
  • What is the latent space of the QINN, the quantum base distribution?

I'm looking forward to your resubmission! Happy holidays! And sorry for the late report. Somehow, I missed the reminder I got in November.

Requested changes

see report

  • validity: top
  • significance: top
  • originality: top
  • clarity: good
  • formatting: excellent
  • grammar: excellent

Author:  Armand Rousselot  on 2024-01-08  [id 4229]

(in reply to Report 3 on 2023-12-22)
Category:
answer to question

Thank you for your questions and feedback. We are happy to clarify a few points about the QINN. First, note the INN and QINN map between the same real-valued data and latent space. The only difference is that in the QINN, this mapping goes through quantum states. At the end of the QINN circuit, we perform a measurement, which results in real-valued outputs that are no longer quantum states. The advantage of the QINN is not that it lets us map the data to a quantum distribution (although this would be an interesting idea to explore) but that it lets us map the data to a real-valued latent distribution *more efficiently*.

Thank you for pointing out the missing discussion of the differences and similarities between the INN and the QINN. The QINN does indeed learn a parameterized transformation between samples of two distributions analogous to the INN. Both minimize the negative log-likelihood and enable sampling by invertibility. The difference is that the function is now represented by a QNN instead of a coupling-based classical INN. Coupling-based INNs require substantial architectural restrictions to ensure the tractability of the equation (7). We show that these architectural restrictions are not necessary to impose on a QNN for density estimation; instead, a simple auxiliary loss to ensure invertibility suffices in theory.

In practice, we observe that we require an additional MMD loss to ensure proper convergence of the model. A short explanation is that the NLL loss only aims at the correct latent/data distribution when the model is invertible, while the MMD does not require this property. Thus, at the beginning of the training, the MMD can have a guiding effect on the model. We will add a full explanation and an overview of the entire loss function to Section 3.

While the train/val/test split might look unusual, the task is comparatively low in complexity, and the model size is tiny. Therefore, the training set size is sufficient to prevent overfitting and describe the data distribution. While the choice to allocate the leftover events to the test set instead of dropping them from the dataset entirely was admittedly quite arbitrary, it should not invalidate any of the results presented in this work.

The poor performance of the INN compared to works like 2110.13632 stems from (intentional) underparameterization, as you correctly pointed out. Since the simulation of quantum circuits is still costly to compute, we have to artificially limit model sizes and task complexities to compare them to classical models. Only when the models are at the limit of being able to express the data distribution can we make observations about the differences in their capacity.

---

## Editorial Decision

published